# Mutational signature analyses in multi-child families reveal sources of age-related increases in human germline mutations
Habiballah Shojaeisaadi [1], Andrew Schoenrock[1,5], Matthew J. Meier [1], Andrew Williams [1], Jill M. Norris [2], Nicholette D. Palmer [3], Carole L. Yauk [4] & Francesco Marchetti [1] ✉

Whole-genome sequencing studies of parent–offspring trios have provided valuable insights into the potential impact of de novo mutations (DNMs) on human health and disease. However, the molecular mechanisms that drive DNMs are unclear. Studies with multi-child families can provide important insight into the causes of inter-family variability in DNM rates but they are highly limited. We characterized 2479 de novo single nucleotide variants (SNVs) in 13 multi-child families of Mexican-American ethnicity. We observed a strong paternal age effect on validated de novo SNVs with extensive inter-family variability in the yearly rate of increase. Children of older fathers showed more C > T transitions at CpG sites than children from younger fathers. Validated SNVs were examined against one cancer (COSMIC) and two non-cancer (human germline and CRISPR-Cas 9 knockout of human DNA repair genes) mutational signature databases. These analyses suggest that inaccurate DNA mismatch repair during repair initiation and excision processes, along with DNA damage and replication errors, are major sources of human germline de novo SNVs. Our findings provide important information for understanding the potential sources of human germline de novo SNVs and the critical role of DNA mismatch repair in their genesis.

Family-based whole-genome or whole-exome sequencing studies predominantly from trios, i.e., parents and a single child, have enabled the identification of de novo mutations (DNMs) in humans[1,2]. DNMs are novel changes in the DNA sequence of an individual that are not present in the parents. DNMs can appear during gametogenesis, post-zygotically, or during the postnatal life of the individual; however, only DNMs that are present in the parental germ cells (i.e., germline DNMs) will be passed on to the next generation and affect all cells in the offspring. Improving our understanding of the determinants of germline DNMs is critical because they are drivers of evolution and human genetic disease[3].

Germline DNMs play a major role in common neurodevelopmental and psychiatric disorders such as intellectual disability, autism[4], schizophrenia[5], and other diseases[6]. Developmental disorders caused by DNMs have a prevalence of 1 in 200–500 births corresponding to ~400,000 affected children born globally per year[7]. Approximately 80% of transmitted DNMs arise in the paternal germline[2,8,9]. Sequencing studies have demonstrated that germline DNMs increase steadily with the age of the father at conception[2,3] and this association is referred to as the paternal age effect (PAE)[10].

Studies of human trio cohorts representing diverse populations and ancestries have reported significant variation in both mutation rate and PAE among human populations and that this variation is not heritable[11]. This suggests that the environment may affect mutation rates more significantly than previously thought[11–13]. For example, a study on an Amish population, which experiences lower exposure levels of environmental contaminants than populations in urban settings, showed a lower mutation rate and PAE than other human populations[11]. The few studies that have sequenced multi-child families have reported high variability in

[1]Environmental Health Science and Research Bureau, Health Canada, Ottawa, ON, Canada. [2]Department of Epidemiology, Colorado School of Public Health, University of Colorado Anschutz Medical Campus, Aurora, CO, USA. [3]Department of Biochemistry, Wake Forest School of Medicine, Winston-Salem, NC, USA. [4]Department of Biology, University of Ottawa, Ottawa, ON, Canada. [5]Present address: Research Computing Services, Carleton University, Ottawa, ON, Canada. ✉e-mail: francesco.marchetti@hc-sc.gc.ca

the rate of DNMs, even among families within the same population[8,14,15]. These intriguing findings emphasize the need to study DNMs from more diverse populations to determine the factors contributing to differences in the rate and spectrum of DNMs within and between human populations.

Recent advances in cancer mutational signature analyses have revealed both endogenous and exogenous sources of mutagenesis in tumors and provided new insights into factors that influence cancer development[16]. Mutational signatures are distinct patterns of mutation accumulation that reflect a combination of cellular processes, such as DNA replication errors caused by endogenous factors, DNA repair deficiencies, and/or exposure to exogenous/environmental mutagens[17]. Comparison of mutation profiles against known signatures, some of which have proposed etiologies/annotations, can provide insight into the potential underlying mutational mechanisms within an observed catalog of mutations[3,16,18]. However, mutational signature analyses have not been widely applied to germline DNMs, and the molecular mechanisms involved in their genesis are still largely unknown.

While the analysis of human germline DNMs with cancer-relevant mutational signatures is informative, the use of more targeted mutational signatures relevant to human germline DNMs may offer a more precise assessment of their potential etiology and underlying mechanisms. Recently, Seplyarskiy et al.[19] identified 14 distinct human germline mutation patterns (originally named Component 1–14) corresponding to nine processes: five DNA strand-dependent (represented by two components each) and four DNA strand-independent. The authors provided a biological interpretation for seven of these processes and found that they explained the variation in mutation properties between loci[19]. Thus, these human germline signatures represent a yet unexplored and critical resource to investigate the mechanisms of human DNMs.

An important role for DNA mismatch repair (MMR) in the genesis of human DNMs has been inferred from the high occurrence of mutations at CpG sites due to spontaneous deamination of 5-methylcytosine (5mC)[20,21]. However, mechanistic studies investigating which steps of the MMR pathway are more critical for the formation of DNMs are lacking. Recently, targeted CRISPR-Cas9-based knockouts (KO) of DNA repair genes in isogenic human induced pluripotent stem cells (hiPSCs) cell lines have generated a dataset of nine DNA repair-deficient mutational signatures, including six MMR genes[22]. The availability of these novel DNA repair KO signatures provides an opportunity to better define the critical steps within the MMR pathway that are involved in the genesis of human DNMs.

In this study, we combined the identification of de novo single nucleotide variants (SNVs) in multi-child human families and exploited three mutational signature databases to characterize inter-family variability in the PAE and the molecular mechanisms of germline mutations. We used whole-genome sequencing analyses of 13 multi-sibling families of Mexican-American ethnicity from the Insulin Resistance Atherosclerosis Family Study (IRASFS)[23] to investigate the PAE in this growing minority population in the USA. Then, we used three existing mutational signature datasets to identify signatures that explained the observed de novo SNV spectrum: (1) the cancer-derived Catalogue of Somatic Mutations in Cancer (COSMIC, v3.3) composed of 60 Single Base Substitutions (SBS) signatures with both known and unknown etiologies[17]; (2) the germline-specific dataset comprising 14 SBS signatures with proposed etiologies[19]; and (3) the dataset from targeted CRISPR-Cas9-based KO of DNA repair genes in hiPSCs cell lines with nine SBS mutational signatures[22]. The integration of the PAE analyses and mutational signatures from three different signature databases allowed us to identify several types of DNA damage and inaccurate MMR as major contributors to the formation of SNVs and their accumulation with paternal age. These findings expand our understanding of the potential sources of human germline de novo SNVs and the critical role of DNA MMR in their genesis as a function of paternal age.

## Results

### Identification and validation of SNVs

Thirteen multi-child families were selected from the Mexican-American population of the IRASFS cohort[24]. On average, selected families had ~4 children (mean ± SD: 3.7 ± 1.2) with a mean paternal age of 27.5 ± 6.4 years at the time of birth with a minimum and maximum age of 16.4 and 41.2 years old, respectively (Fig. 1). Average paternal age difference between the first and last child was 9.3 ± 4.9 years (Supplementary Table 1). After quality control, we sequenced the genomes of 74 individuals, including 26 parents and 48 probands, to a genome-wide median depth of ~30X. Here, we focused our analyses on de novo SNVs (hereafter referred to as SNVs).

We used two distinct variant calling software tools to maximize the identification of candidate SNVs (Supplementary Fig. S1): DeNovoGear and GATK. DeNovoGear identified 123–387 candidate SNVs per child (average: 237.6); while GATK identified 24–111 candidate SNVs per child (average: 57.2). In total, 11,403 and 2729 SNVs were identified by DeNovoGear and GATK, respectively. Over 90% of the GATK-identified SNVs overlapped with those of DeNovoGear (Supplementary Fig. S2A), which generated an overall list of 11,590 candidate SNVs. Among these, ~600 SNVs that were observed more than once among all children were eliminated from further analyses. Following targeted resequencing of 6118 candidate SNVs with successfully designed baits, 2479 SNVs were validated (Supplementary Fig. S2B). This resulted in an average germline mutation rate of $1.03 \times 10^{-8}$ (95% CI: $0.96 \times 10^{-8}$–$1.1 \times 10^{-8}$) per base pair per generation. We found an average (mean ± SD) of 51.6 ± 11.7 (range: 29–82) validated SNVs per proband, which is in line with other published studies (Supplementary Fig. S3), and 190.7 ± 78.6 (range: 88–342) validated SNVs per family.

### There is extensive inter-family variability in the PAE

Analysis of validated SNVs demonstrated a strong PAE overall. SNVs increased with an estimated slope of 1.29 (95% CI: 0.83–1.74, $p < 0.0001$) SNVs for each additional year of the father's age at the time of child's birth (Fig. 2). Analyses of individual families showed a wide range of estimated confidence intervals surrounding the slope point from an average of nearly no change, i.e., 0.03 (95% CI: −0.1–0.2; IRASFS Family 03) to more than 6.52 (95% CI: 5.5–7.5; IRASFS Family 05) additional SNVs for each increasing year of paternal age (Fig. 3). Interestingly, we observed one family with four offspring (IRASFS Family 09) that had a negative PAE. In fact, the number of validated SNVs in this family decreased from 51 to 41 from the first child (p1; paternal age: 16.4 years) to the last child (p4; paternal age: 22.3 years), respectively, which resulted in a negative slope of −1.88 (95% CI: −2.3–−1.4). Together, our analyses demonstrate extensive inter-family variability in the PAE in this cohort.

The set of 2542 candidate SNVs that were identified by both DeNovoGear and GATK also showed a consistent PAE and inter-family variability in the slope of increase (Supplementary Fig. S4). Furthermore, when sorted by the slope of increase, the bottom three (e.g., IRASFS Families 09, 03, and 02) and top three (e.g., IRASFS Families 04, 01, and 05) families were the same as when the validated SNVs were used. However, since not all these candidate SNVs could be re-sequenced and validated, separate data analyses on this dataset are not presented.

### The majority of the validated de novo SNVs have a paternal origin

We investigated the parental origin of validated SNVs using read-based phasing and a haplotype assembly approach. Additionally, we visually verified the phasing result of each SNV using the Integrative Genomics Viewer (IGV) to ensure accuracy. We determined the parental origin for an average (mean ± SD) of 10.4 ± 1.7% validated autosomal SNVs per IRASFS family (range: 7.1–13.2%). As expected, this analysis identified a significant male bias in the contribution of SNVs with a 5.4:1 ratio of validated paternal:maternal autosomal SNVs, and a mean of 78.6% (95% CI: 71.7%–85.6%) of autosomal SNVs with paternal origin (Fig. 4A). For one family (IRASFS Family 04), all phased SNVs were of paternal origin. Overall,

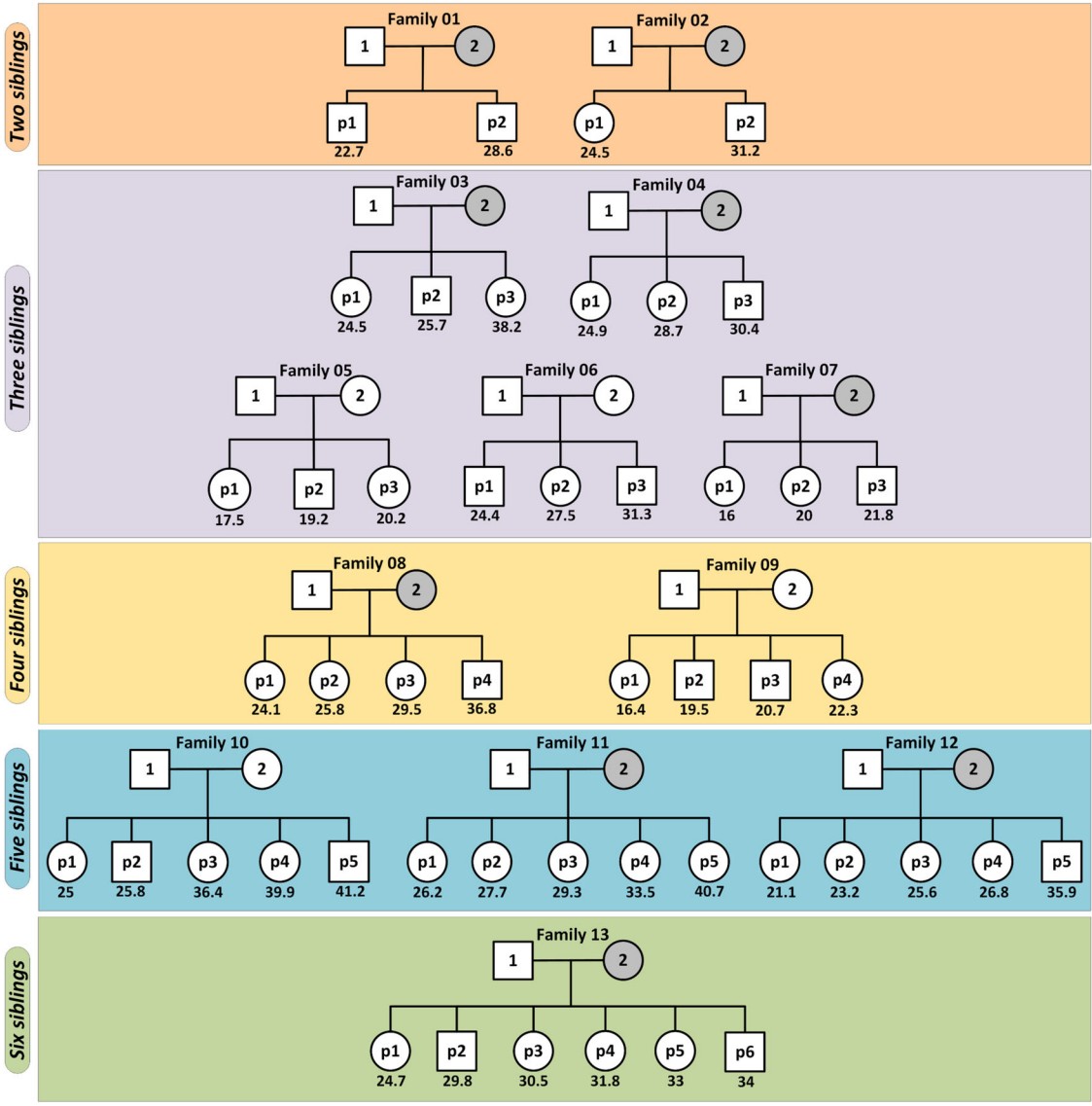

**Fig. 1 | Pedigrees of the 13 multi-child IRASFS families.** Each IRASFS Family is labeled with a sequential number 1–13. Within each family, fathers and mothers are identified with the numbers 1 and 2, respectively, while each child is identified by the letter p (proband) and a number representing the order of birth. The number under each child represents the paternal age at the time of birth for that child. The youngest and oldest paternal ages in this IRASFS cohort are 16.4 years old (IRASFS Family 9 - proband #1) and 41.2 years old (IRASFS Family 10 -proband #5), respectively. Gray circles indicate the individuals that were sequenced for this study. All other whole-genome sequences were already available from IRASFS.

we observed a PAE even when considering phased SNVs of paternal origin only (Fig. 4B).

**Mutation signature analyses identify mutational processes contributing to de novo SNVs**

Next, we analyzed the types of mutations that contributed to the validated SNVs. In agreement with previous studies, transitions were more common than transversions. The most common SNVs were C > T transitions, particularly at all four CpG sites (Fig. 5). ACG trinucleotides had the highest numbers of mutations, followed by CCG, GCG and TCG, respectively. T > C transitions were the next most common mutations, especially within the ApTpN trinucleotide context (i.e., ATA, ATG, and ATT). T > A transversions were the least common mutations. We then separated the validated SNVs into quartiles based on paternal age at the time of child's birth and generated a 96-trinucleotide spectrum for the children born from the youngest and oldest fathers, below 24 and above 33.1 years of age, respectively. A comparison of these spectra showed that the most apparent

difference was an increase in C > T mutations at CpG sites, especially at the CCG and TCG motifs, in children born from the oldest fathers (Fig. 5).

To explore the mutational processes involved in the formation of SNVs, we first performed de novo signature extraction to identify the mutational signature within our dataset. Next, to delineate potential underlying mechanisms, we performed decomposition and fitting analyses on the extracted signature with three published mutational signature datasets starting with the COSMIC signatures.

The COSMIC analysis showed that the two known clock-like age-relevant mutational signatures, SBS1 and SBS5, were the only two signatures needed to explain the observed pattern of de novo SNVs. In fact, decomposition of the extracted SNV signature showed that a combination of 85% SBS5 and 15% SBS1 generated a reconstructed signature with a cosine similarity value of 0.989 with the extracted one (Fig. 6A). SBS1 is due to spontaneous deamination of 5mC, while SBS5 has an unknown etiology (Fig. 6A). Repeating the analysis using the recently expanded repertoire of cancer mutational signatures from the Genomics England Limited (GEL

**Fig. 2 | The distribution of validated de novo SNVs in the IRASFS multi-child families and their correlation with paternal age.** The scatter plot represents the number of validated de novo SNVs in each of the 48 children by paternal age at the time of birth. Each color represents a specific IRASFS family. The red line represents the slope of all validated de novo SNVs and the shaded area is the 95% confidence interval for the regression line.

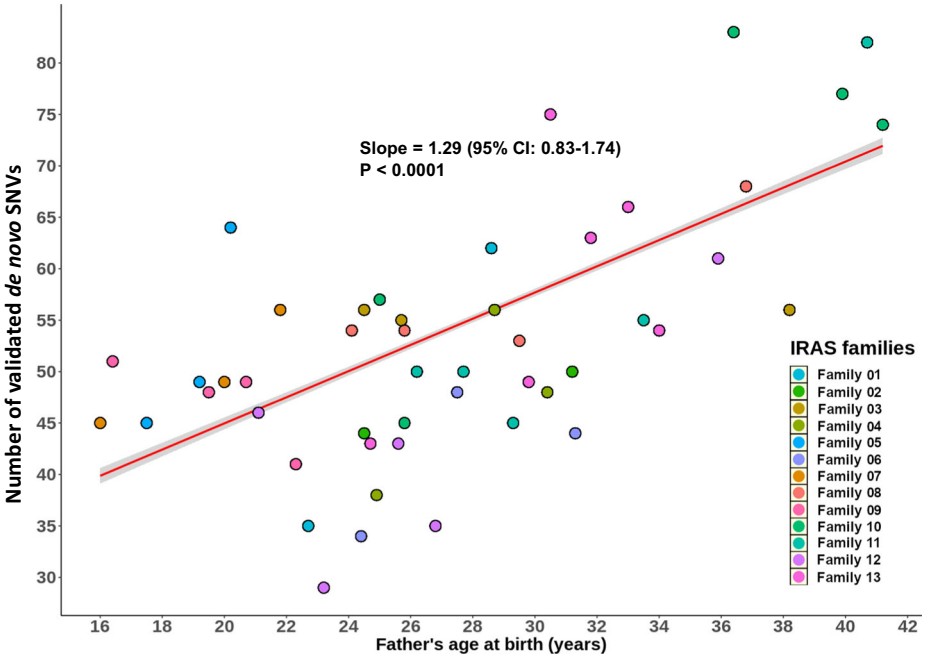

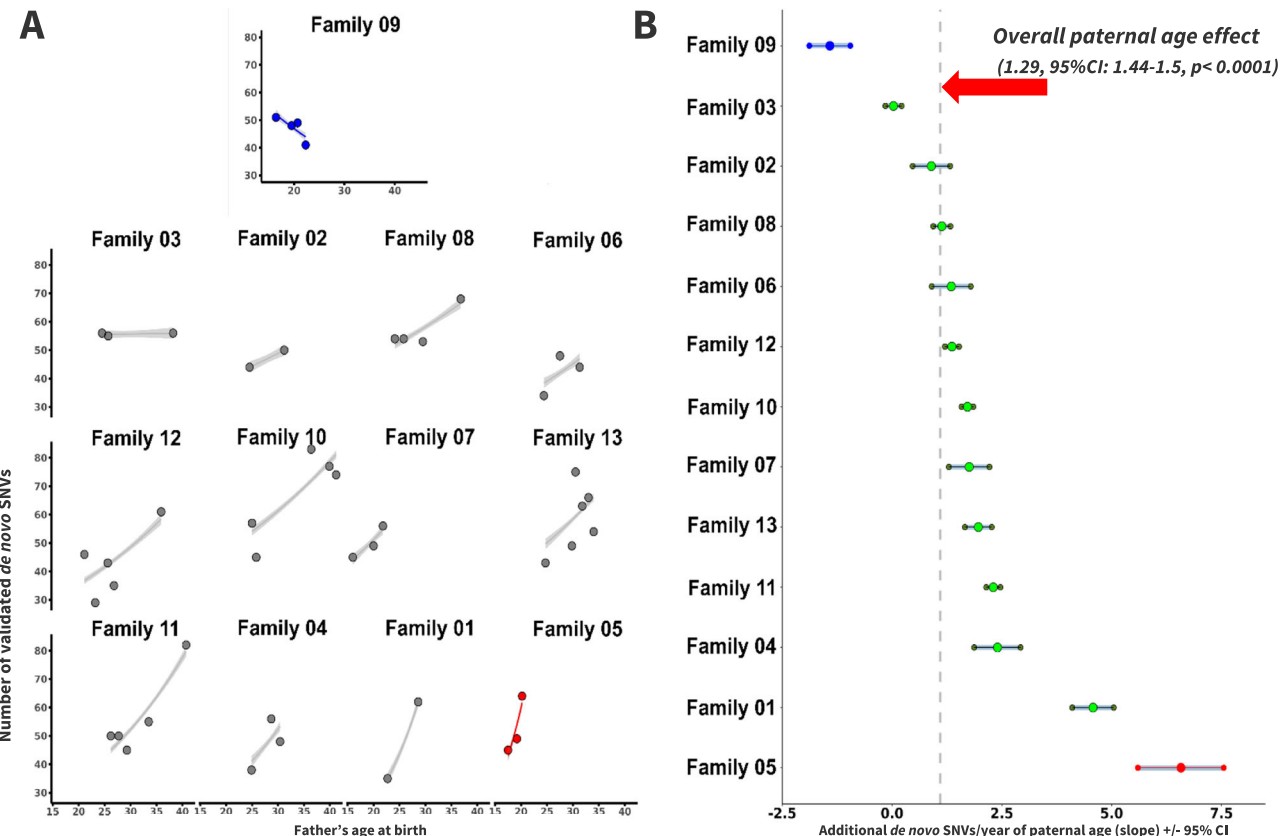

**Fig. 3 | Inter-family variability of the paternal age effect among the IRASFS multi-child families. A** Scatter plots of the numbers of de novo SNVs for each family relative to the father's age at each child's birth, ordered by slope from the lowest (top with blue color, IRASFS Family 09) to the highest rate (bottom right with red color, IRASFS Family 05). Regression lines and 95% confidence intervals indicate the predicted number of de novo SNVs as a function of paternal age using a Poisson regression. **B** Slope ± 95% confidence interval (CI) of each IRASFS family sorted in order of increasing slope as in **A**. The dashed vertical line indicates the paternal age effect based on the combined data from all families (1.29 de novo SNVs/year, 95%CI: 1.44–1.57, $p < 0.0001$).

**Fig. 4 | The parent of origin of de novo SNVs in the IRASFS families. A** Box plots representing the proportion of validated de novo SNVs that were successfully phased to establish the parent of origin. **B** Scatter plot with fitted regression line ± 95% confidence interval of the distribution of phased de novo SNVs in each of the 48 children based on the paternal age at the time of birth. Maternally-based SNVs are plotted according to the age of the father at the time of child's birth because maternal age was not available.

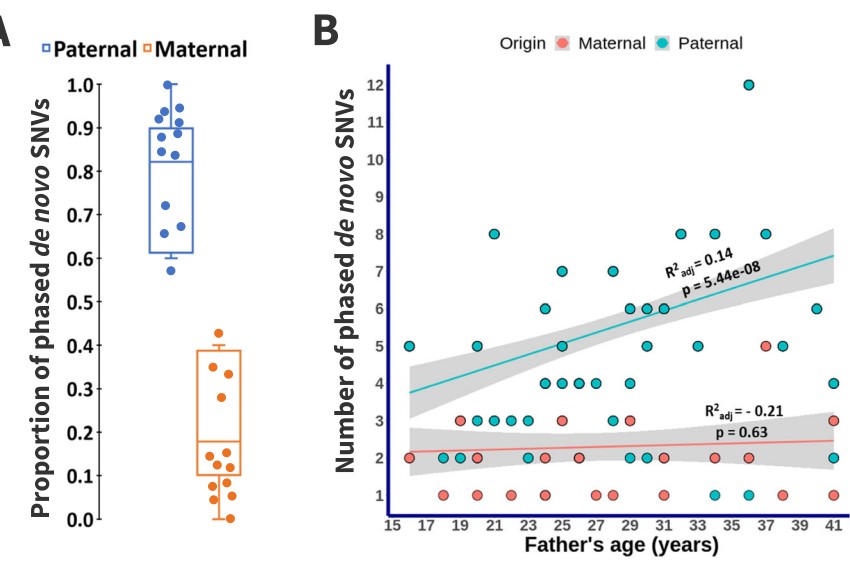

2022)[25] did not change the outcome. In fact, a combination of ~73% SBS5 and ~27% SBS1 generated a reconstructed signature with a cosine similarity value of 0.972 with the extracted one.

Next, we investigated how the IRASFS-extracted mutational signature could be decomposed and fitted using the human germline mutational signatures[19]. This analysis showed that three human germline mutational patterns, identified as human germline component 1, 3, and 10, generated a reconstructed signature with a cosine similarity value of 0.954 with the extracted one (Fig. 6B). Proposed mechanisms for these three components are: asymmetric resolution of bulky DNA damage (component 1); replication errors (component 3); and 5mC deamination or erroneous replication over methylcytosine (component 10) (Fig. 6B). Our analysis revealed that component 1, characterized predominantly by T > C transitions, accounted for the highest proportion (~45%) of the SNV mutation pattern. Component 10, which is characterized by C > T transitions at CpG motifs (i.e., NpCpG; N = A, T, C, G) contributed ~37%, while component 3, characterized by C > T transitions with no enrichment for a specific motif, contributed ~18%.

To understand the origin of the bulky DNA damage suggested by component 1, we compared the SNV mutation signature individually against the compendium of mutational signatures of environmental chemicals in hiPSCs[26]. This analysis showed that our SNV mutational signature had the highest cosine similarity values with dimethyl sulfate (0.552), an alkylating agent, and dibenzo[a,l]pyrene-diol-epoxide (0.527), a polycyclic aromatic hydrocarbon that forms bulky DNA adducts.

We then compared the extracted SNV signature to nine SBS signatures derived from human DNA repair gene KOs (i.e., ΔEXO1, ΔMLH1, ΔMSH2, ΔMSH6, ΔOGG1, ΔPMS1, ΔPMS2, ΔRNF168 & ΔUNG) plus the background (i.e., control hiPSCs without any KO) generated in a comprehensive CRISPR-Cas9-based KO in hiPSCs isogenic cell lines[22]. Our extracted SNV signature could be decomposed and best fitted with a combination of three human DNA MMR repair genes: ΔEXO1 (~41%), ΔPMS1 (~35%) and ΔPMS2 (~23%). The reconstructed signature had a cosine similarity value of 0.915 with the extracted one (Fig. 6C). The ΔEXO1 signature is identified by relatively high T > C transitions, especially at ATA and TTA motifs. The ΔPMS1 signature is characterized predominantly by C > T transitions, particularly at NpCpG sites, as well as its ACA motif. The ΔPMS2 signature is predominantly composed of T > C transitions, especially at ATA, ATG, and CTG trinucleotides (Fig. 6C).

Finally, we attempted signature extraction, decomposition and fitting using the two mutation spectra for the children born from the youngest (Fig. 5B) and oldest (Fig. 5C) fathers to explore whether the contribution of the identified signatures changed with paternal age. However, this analysis

generated reconstructed signatures with much lower cosine similarity values with the extracted signature than when the entire set of SNVs was used (Supplementary Table 2). This finding was consistent for all three mutational signature datasets. We interpret these results to mean that the separation of the SNVs in quartiles resulted in an insufficient number of mutations for robust signature extraction and reconstruction.

## Discussion

We validated ~2500 de novo SNVs in 13 muti-child families with Mexican-American ethnicity from the IRASFS cohort and provided possible mechanisms for the genesis of human germline SNVs. We observed a strong PAE with extensive inter-family variability and, as expected, the majority of SNVs had a paternal origin. In addition, we found that C > T transitions at CpG sites were more common in children from older fathers. Our signature analyses suggest that SNVs originated from several molecular mutagenic processes, including deamination of 5mC, bulky DNA damage, replication errors, and inaccurate MMR. Finally, we propose a model identifying the critical role of DNA MMR in the genesis of SNVs as a function of paternal age.

The analysis of diverse populations and ethnicities in genetic studies[27–29] is of great importance to identify factors that determine differences in susceptibility to genetic disorders (e.g., asthma, cancer, diabetes, and atherosclerosis), responses to intervention therapies[30], and environmental exposures (e.g., air pollution[31]). We found that de novo SNVs exhibit a strong PAE with significant variation among Mexican-American families that are aligned with two previous studies of different ethnicities (i.e. CEPH/Utah cohort of white-American ethnicity[14] and Middle Eastern families with heterogeneous ethnicity[15]). In addition, when the multi-child families from these two previous studies and ours are sorted based on the increasing slope of the PAE, we observed a random distribution of the families, irrespective of ethnicity (Supplementary Fig. S5). Therefore, it appears that the PAE and its inter-family variability is a general characteristic of the human species that is independent of ethnicity. Inter-individual variation in DNA replication error rates, DNA repair efficiencies, and endogenous and exogenous sources of DNA damaging compounds are likely the major determinants of the observed variability in the PAE within and across diverse human populations[19,32,33].

To the best of our knowledge, we report the first instance of a family (IRASFS Family 09) with a negative slope of −1.88 for the PAE. We believe that this apparently unexpected observation is not indicative of a fundamental biological difference in this father but is a consequence of his young age (22.3 years old at the time of the fourth child's birth). In fact, the number of validated SNVs observed in the four children is consistent with the range

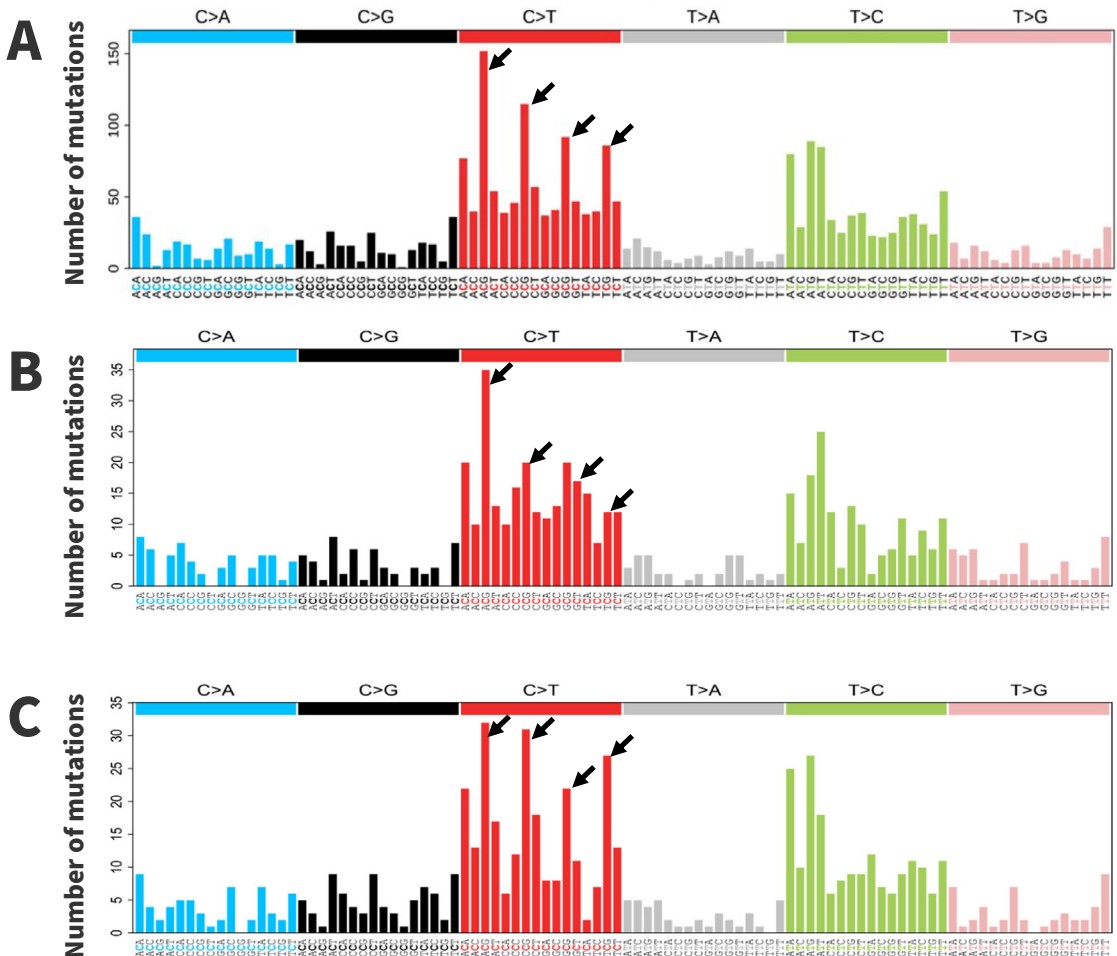

**Fig. 5 | The 96-trinucleotide mutation spectrum of de novo SNVs in the IRASFS cohort.** Spectra are presented for: **A** de novo SNVs identified in all children from this study; **B** children born from the youngest fathers (<24 years of age); and, **C** children born from the oldest fathers (>33.1 years of age). For the analyses shown in **B**, **C**, children were separated into quartiles based on the age of the fathers. Data are presented as total counts of SNVs. Arrows indicate CpG sites.

of SNVs observed in fathers of similar age (Supplementary Fig. S3). Secondly, both previous multi-family studies[14,15] have several cases with downward trends in the number of SNVs when limited to the first few children. It is only because these families have children fathered at an older age that the slope turned positive. Thus, we hypothesize that if IRASFS Family 09 had a fifth or sixth child, this negative slope would disappear. Overall, these data support the role of stochastic factors in the SNV mutation rate and PAE variation[34].

In agreement with previous studies, we found that ~80% of the de novo SNVs originated from the paternal genome. This has long been attributed to DNA replication errors occurring more frequently in the male germline due to the higher number of cell divisions with respect to the female germline. DNA replication errors have been considered the predominant source of germline mutations[3,20]. However, recent human evidence points to the importance of mutagenic processes that do not depend on cell division and suggests that other mechanisms, such as sex-based differences in endogenous sources of DNA damage or DNA repair mechanisms, are also contributing to the preferential generation of SNVs in the paternal germline[35,36].

The observed SNV mutation spectrum is the result of multiple mutation mechanisms operating in the germ cells of the parents. Thus, interrogation of its characteristics provides clues to its origin. The SNV mutation spectrum in this study was characterized by high frequencies of C > T transitions, which is the most frequent mutation in human populations[20] accounting for one-third of the SNVs responsible for hereditary diseases[37]. The occurrence of C > T transitions, particularly at CpG

sites, immediately suggests spontaneous deamination of 5mC as the likely culprit. The high mutagenicity of cytosines at CpG sites with respect to any other nucleotide in the human genome is well known[21]. Cytosines in CpG dinucleotides are often methylated. Spontaneous deamination of 5mC generates thymine, while spontaneous deamination of unmethylated cytosine produces uracil. Deaminated 5mC is less efficiently repaired prior to DNA replication[38] by the MMR repair machinery[21,39] than uracil, which is more efficiently repaired by base excision repair (BER)[40]. Thus, spontaneous deamination of 5mC is more likely to result in a C > T transition at CpG sequences[39]. Our findings are in agreement with studies in different ethnicities[3,8,9,14,15,20,38] demonstrating that this specific mutational pattern appears to be independent of the human population background[41]. Furthermore, the comparison of the 96-trinucleotide mutation spectra for the children born from the youngest and oldest fathers in our IRASFS families suggests that C > T mutations at CpG sites increase with paternal age.

A few studies[8,12] have used mutational signature analyses of de novo SNVs obtained from human families; these studies limited their analyses to COSMIC signatures to reconstruct the observed mutation spectrum. Furthermore, Kaplanis et al.[12] conducted mutational signature analyses exclusively on those trios with a hypermutator phenotype caused by preconceptional paternal exposure to chemotherapy or because of DNA repair defects. We have expanded on these studies by implementing a systematic signature analysis using multiple signature databases to obtain further insight into the mechanisms underpinning de novo SNV formation. Using the COSMIC database[17,25], we found that SBS1 and SBS5 are the two mutational signatures that best reconstructed the pattern of de novo SNVs

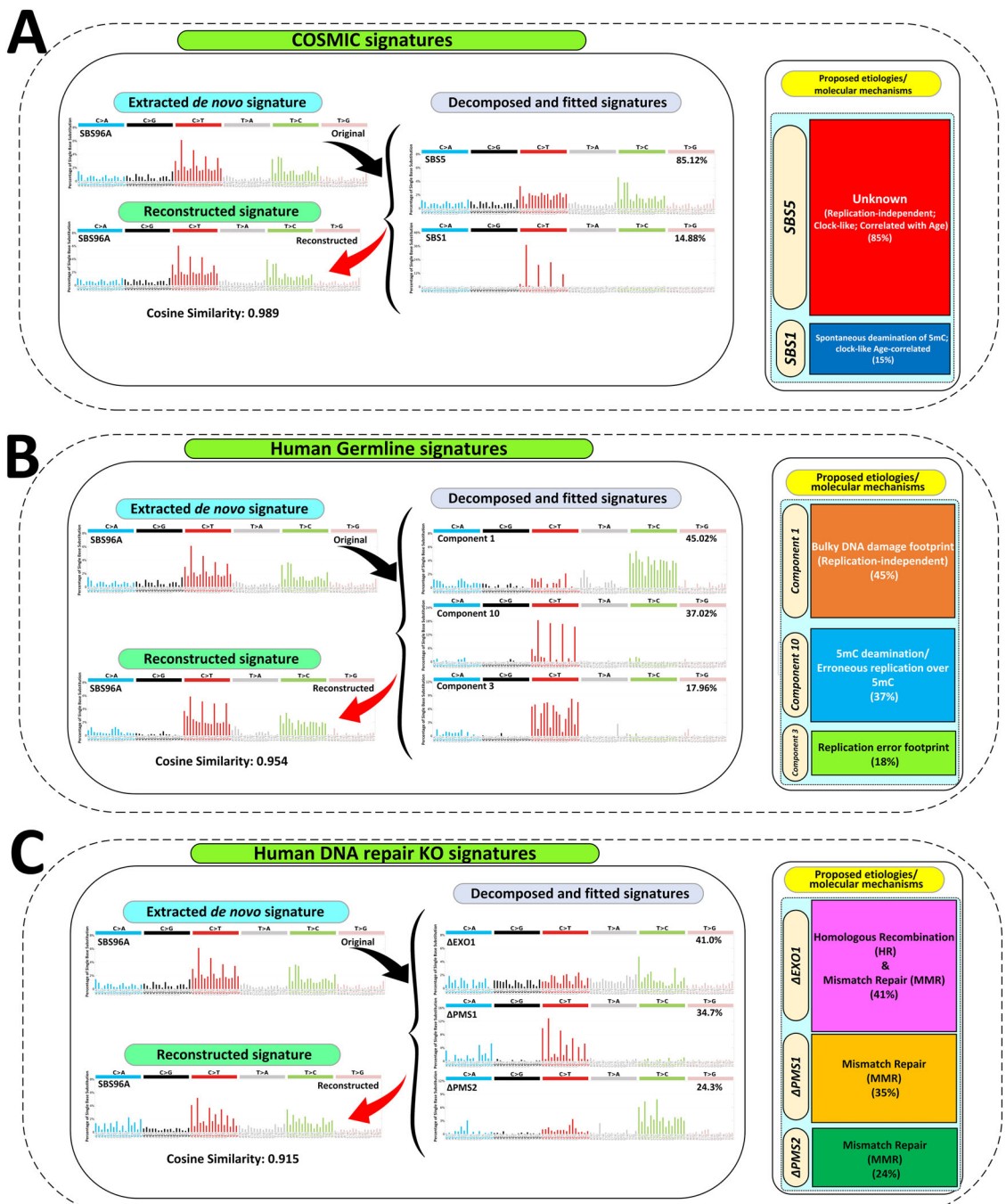

**Fig. 6 | Mutational signature analyses of validated de novo SNVs in IRASFS families.** Decomposition and fitting analyses of the de novo SBS mutational signature (SBS96A Original) extracted by SigProfilerExtractor from the validated de novo SNVs using: **A** COSMIC SBS signatures; **B** human germline mutational signatures; and **C** mutational signatures from targeted CRISPR-Cas9 Knockouts of DNA repair genes in human iPSCs. For each mutational signature dataset, the left panel shows the extracted signature, the fitted signatures with their percent contribution, and the reconstructed signature with the cosine similarity value; the tile plot on the right reports a visual representation of the identified proportion of each signature and its etiology/annotation.

(cosine similarity = 0.99), as expected[3,9,38]. A combination of SBS1 and SBS5 is thought to contribute to mutation accumulation with age in most normal human somatic and germline cells[36,42–44]. Both SBS1 and SBS5 correlate with the patient's age in many cancer types and are known as clock-like signatures[38]. However, they have different etiologies[18,36,38]. The mutational process underlying SBS1 is the deamination of 5mC at CpG dinucleotides[45]. SBS1 is largely cell division dependent and is strongly associated with late-replicating DNA, either because the mutagenic process is more active at this time or because of reduced activity of replication-coupled repair mechanisms in late-replicating DNA[46]. In contrast, SBS5 has an unknown etiology

and is independent of cell proliferation rate[38]. SBS5 possesses replication- and cell-cycle-independent characteristics[45,47], and its mutational pattern appears to be driven by exogenous factors accumulating over time, such as continuous exposure to reactive oxidative species[38,42,48]. Overall, the COSMIC signatures identified age-related deamination of 5mC and replication-independent processes as major contributors to de novo SNVs.

Fitting the human germline mutational signatures[19] to our data demonstrated that the SNV signature is best reconstructed (cosine similarity = 0.95) by the human germline components 1, 10, and 3 in decreasing proportions. Component 1 is replication-independent, strand-dependent,

**Fig. 7 | Proposed model for human germline de novo SNV formation with increasing paternal age.** DNA damage incurred from environmental exposures and cellular processes associated with normal physiological processes and aging are shown at the top. The molecular signatures identified by querying the three signature databases are shown below, the type of DNA damage. Arrows connect the damage/signatures to the DNA repair pathway that is responsible for repairing that specific DNA damage. (see discussion for a full description). Dashed arrows indicate the processes that are suggested to be involved based on these analyses. **5mC** 5-methyl-cytosine, **EXO1** Exonuclease, **MMR** mismatch repair pathway, **NER** Nucleotide excision repair pathway, **PMS1** PMS1 Homolog 1, Mismatch Repair System Component, **PMS2** PMS1 Homolog 2, Mismatch Repair System Component, **SBS1** COSMIC single base substitution signature 1, **SBS5** COSMIC single base substitution signature 5.

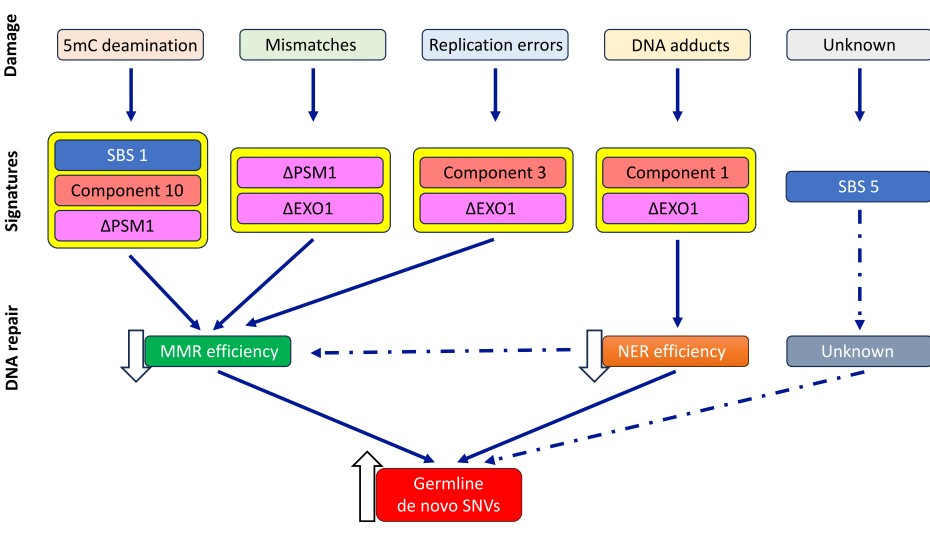

correlates with experimentally obtained transcription-coupled repair (TCR) activity[49], and is thought to be the footprint of asymmetric resolution of bulky DNA damage[19]. Component 10 is characterized by CpG transitions mediated by 5mC deamination or by erroneous replication over 5mC. Lastly, Component 3 is the footprint of replication errors. While the contributions of 5mC deamination and replication errors were expected, the implication of bulky DNA damage in the genesis of a large portion of de novo SNVs, as suggested by component 1, is striking for two reasons. First, it provides additional support to the growing line of evidence that mutational processes that are independent of cell division are important contributors not only to somatic cell mutagenesis[45,47] but also to human germline mutagenesis[18,33,50,51]. Second, it suggests a critical role for exogenous exposures in the genesis of de novo SNVs and provides support for the notion that DNA damage, in addition to DNA replication and cell division[2,3,20,52], is an underappreciated source of new germline mutations[18,33,35]. Thus, the use of the human germline signatures suggests that germline mutations in males are not simply due to more cell divisions but also due to a different balance of DNA damage versus DNA repair.

An important role for DNA damage and repair deficiency in germline mutagenesis is emerging[18,33]. To address this fundamental aspect, we leveraged the recent CRISPR screening using hiPSCs that identified SBS signatures originating from KO of human DNA repair genes[22]. This approach enabled us to identify a critical role for three MMR genes (*PMS1*, *PMS2*, and *EXO1*) in human germline mutagenesis. We found that the combinatorial signature resulting from these three MMR KO genes best recapitulated (cosine similarity >0.9) the pattern of de novo SNVs extracted from the IRASFS cohort. This result is aligned with the COSMIC and the human germline signature analyses that identified an important role of 5mC deamination and replication errors in the genesis of human germline SNVs. Furthermore, this finding is in agreement with several reports indicating that the dominant mutational processes in the germline, whether originating from replication errors or mediated by DNA damage, are expected to produce mismatches[10,33,53,54] and that MMR plays a critical role during meiosis, gamete formation, and germline DNA damage repair[55–57]. Our analysis further suggests that repair initiation and lesion excision are the critical MMR steps involved in the formation of SNVs. In addition, a role for Homologous Recombination repair is suggested since EXO1 is a critical component of this pathway and has pleiotropic roles in DNA repair and replication[58,59].

We summarize the results from the mutation signature analyses in a model describing the molecular mechanisms underlying the age-related increase of de novo SNVs (Fig. 7). Our analyses support a central role for MMR, particularly inefficiency in the initiation and excision steps of the pathway, in the formation of SNVs. This is further supported by the

common identification of 5mC deamination and replication errors, as well as the SBS1 signature, from human germline signature and COSMIC signature analyses, respectively. In addition, the high contribution of the human germline signature with a footprint of bulky DNA damage suggests a role for inadequate nucleotide excision repair (NER) in human germline SNV formation. However, this could not be computationally tested since no NER-associated signatures were available due to the cellular lethality resulting from knocking out human NER genes[22]. Finally, we identified a very high contribution of the age-related SBS5 signature to de novo SNVs. Although its etiology is yet to be elucidated, it shares high similarity with the signature of *ΔEXO1* and *ΔRNF168*, a ubiquitin ligase that functions as a chromatin modifier during DNA damage repair[60] in hiPSCs[22]. Due to the wide-ranging roles played by these proteins, it is likely that SBS5 has a complex etiology and originates from multiple repair pathways that deal with exogenous and endogenous DNA damage. Consistent with other studies[51,61,62], our model shows that replication errors are not the main driver of de novo SNVs. Rather, our model proposes that declines in the efficiency of DNA repair pathways with age[63–65], together with an accumulation of endogenous and exogenous DNA damage, ultimately lead to increases in human de novo SNVs with advancing paternal age.

As in the other two multi-child family studies[14,15], we observed inter-family variation in the PAE, suggesting variation in the underlying mechanisms. We attempted to examine whether mutational signature differences among the IRASFS families contributed to the observed variability in the PAE. These analyses did show some inter-family differences, especially when using the germline and DNA repair KO signatures; however, the reconstructed signatures had average cosine similarity values among the 13 families of 0.892, 0.775, and 0.809 for COSMIC, germline, and DNA repair KOs, respectively (Supplementary Table 2). Overall, 46% of cosine values obtained when using the germline and DNA repair KO signatures were below the threshold (i.e., 0.8) that was considered to occur purely by chance[66], which greatly reduced the confidence in the observed differences.

Variation in human germline mutation spectra has been attributed to population-specific genetic factors or environmental exposures. For example, an increased rate of TCC > TTC mutations in people from Western Eurasia and South Asia was ascribed to differences in the rate or efficiency of repair of deaminated methylated guanine[67]. Therefore, our findings are likely driven by inter-individual variation in endogenous processes and exogenous environmental factors. One possible mechanism for such variation could be epigenetics. It is well documented that epigenetic factors can modulate DNA repair mechanisms[68–70] and alter the footprint of the mutation process in cancers[71–73]. In addition, genomic and epigenomic features such as recombination rate, replication timing, DNase hypersensitivity, GC content, nucleosome occupancy, simple repeats, and the

trinucleotide context can all influence de novo SNVs[74]. Exploration of the role of epigenetics on inter-family variability in the PAE is an area that requires further study.

There are limitations to the signature analyses that we have conducted. First, the majority of the available SBS signatures have proposed etiologies that are yet to be experimentally validated and may not necessarily represent unique mutational processes. Thus, the identification of a specific signature contributing to a mutational pattern does not establish causation between the proposed signature etiology and the observed mutations. Second, we were limited to mutational signatures for the few DNA repair KO genes that were viable in the hiPSCs model[22]. Although it is unlikely that a wider interrogation of DNA repair pathways would have significantly diminished the central role of MMR, it is possible that a role for NER or BER in the genesis of de novo SNVs would have been better defined. Finally, the range of paternal age between the first and last child in our cohort was limited (i.e., ~10 years), which could have impacted our ability to identify changes in the contribution of mutational signatures to the de novo SNVs with increasing paternal age.

## Conclusion

We exploited mutational signatures from both cancer and non-cancer datasets to provide a comprehensive picture of the mechanisms involved in the genesis of human germline SNVs with advancing paternal age. Although some of the conclusions from these signature analyses recapitulate previous findings (e.g., the role of 5mC deamination in mutagenesis), they also provide new insight into the etiology of SNV formation. Specifically, our analyses suggest that an age-related increase in DNA replication errors during spermatogenesis is not sufficient to explain the etiology of de novo SNVs. Rather, accumulation of both endogenous and exogenous DNA damage and inaccurate DNA damage repair mechanisms are potential sources of human germline de novo SNVs that are impacted by paternal age. In particular, our analyses show an important role for bulky DNA damage and inefficiency of the MMR initiation and lesion excision complexes in the formation of SNVs. Our findings suggest that variations in these processes contribute to the extensive inter-family variability of the PAE.

## Methods
### Study cohort
The IRASFS cohort is a population-based cohort designed to investigate the genetic and epidemiologic basis of glucose homeostasis and abdominal adiposity with a focus on Mexican-derived participants[23]. Broadly, Mexican-American families were recruited from two clinical centres including San Antonio, TX, and San Luis Valley, CO, in 1999–2002 as an extension of the original IRAS cohort recruited in 1992–1994[75]. The overall cohort was relatively healthy and devoid of severe Mendelian diseases. Individual-level genetic data and ADMIXTURE analysis indicated homogeneity across the cohort[76,77]. Specific to the 13 families studied here, and mirroring the larger cohort[78,79], subjects were mostly female (64.6%) with an average age of 47 years, overweight (27.20 kg/m$^2$) and with a near-optimal lipid level (104.24 mg/dL).

The use and handling of human samples in this study were approved by the Research Ethics Board of Health Canada and the Public Health Agency of Canada under protocol REB 2016-001H. For the IRASFS cohort, all study protocols were approved by the Institutional Review Board of each participating clinical and analysis site, and all participants provided written informed consent. All ethical regulations relevant to human research participants were followed.

### Whole-genome sequencing, data pre-processing
We studied 13 multi-child families from a Mexican-American population (26 parents and 48 siblings) of the IRASFS cohort[24]. WGS for the majority of the IRASFS individuals were already available[80]. Here, we performed WGS on nine maternal samples (Fig. 1). Briefly, 300 ng of high-quality gDNA were extracted from blood, and libraries were

prepared using TruSeq DNA PCR-Free Library Prep Kit (Illumina Inc, San Diego, CA, USA). The samples were sequenced using the Illumina HiSeq X Ten instrument by Macrogen (Rockville, MD, USA), targeting a mean depth of 30X (paired-end, 150 bp reads), and the raw reads were aligned to GRCh38 reference genome using BWA-MEM v0.7.17, sorted and indexed with SAMtools V1.8. The aligned reads were filtered to remove duplicate reads resulting from clonal amplification of the same fragments during library construction and sequencing using Picard MarkDuplicates. Base quality recalibration and local realignment were carried out using Genome Analysis Toolkit (GATK) (V 4.0.11.0) best practices workflow (Supplementary Fig. S1).

### Identification of candidate de novo SNVs, quality control, and filtering pipeline
We focused on SNVs and did not include indels in our analyses. To identify candidate SNVs from the WGS data, we implemented two distinct computational methods and variant caller software (Supplementary Fig. S1). The first was based on DeNovoGear (V 1.1.1-308-g3ae70ba), a piece of purpose-built software used to detect somatic and germline SNVs, that identified 11,403 candidate SNVs with its default parameters. The second software was GATK (V 4.0.11.0), the industry standard for identifying SNVs and indels in germline DNA, which identified 2729 SNVs; however, over 90% of these SNVs were also detected by DeNovoGear (Supplementary Fig. S2A). First, we removed variants within low-complexity regions or simple repeats based on UCSC genome track browser data. Then, we removed SNVs that had >10% reads in either parent to ensure that the child possessed a unique genotyped allele absent from both parents. Following the high read count filter, we removed SNVs that did not have at least two forward & reverse reads supporting the SNVs. We required the aligned sequencing depth in the child and both parents to be ≥12 reads, Phred-scaled genotype quality (GQ) to be ≥20 in the child and both parents, and no reads supporting the allele in either parent. Among the 11,590 candidate SNVs that resulted from merging the DeNovoGear and GATK dataset, 595 SNVs were identified in several children and were eliminated, resulting in 10,955 unique SNVs (Supplementary Fig. S2B).

### Targeted resequencing of de novo SNVs and quality control and filtering pipeline
The targeted resequencing validation was performed on the unique candidate SNVs ($n = 10,955$). Baits were designed for >55% of these unique candidate SNVs ($n = 6118$) (Supplementary Fig. S2B). Targeted sequencing of the custom panel was designed with SureSelect DNA Design (Agilent Technologies), and resequencing was performed on the pulldown library following SureSelect XT HS low input Target Enrichment System (Agilent Technologies). The library was sequenced using the Illumina HiSeq 4000 platform with a high coverage depth of ~300X (paired-end, 150 bp reads) at McGill University and Génome Québec Innovation Center. The targeted re-sequenced candidate SNVs underwent data pre-processing as described above. The read counts within capture bait targets was calculated on the SNV pre-processed BAM files by SAMtools Mpileup V1.8. The SNVs were called by BCFtools V1.8. Regions 5 bp up/downstream of de novo SNVs calls were summarized using GATK V4.0.11.0 VariantsToTable to collect depth metrics for reference and alternate alleles before further data processing in R. SNVs with a parental alternate allele fraction (AAF) > 10% were excluded. Furthermore, the remaining candidate SNVs were retained if they met the following criteria: AAF in the proband >0.3 and read depth >10 (Supplementary Fig. S1).

### Identification of parent-of-origin
The main phasing analysis was performed with Unfazed[81] (https://github.com/jbelyeu/unfazed), which applies a novel extended read-based phasing method to determine the parental gamete of origin of SNVs from paired-end Illumina DNA sequencing reads. Unfazed uses variant information for a sequenced trio to identify the parental gamete of origin by linking phase-

informative inherited variants to mutations using read-based phasing. Additionally, WhatsHap[82], a read-based phasing for long reads, was used to complement our phasing results. All phased SNVs were visually validated with IGV.

### SBS mutational signature analyses of de novo SNVs

We applied de novo extraction decomposition and refitting using SigProfiler tools[18]. Initially, the 96-trinucleotide matrix of counts of SNVs was generated by SigprofilerMatrixGenerator v1.1. under default parameters but using hg38 (GRCh38). For de novo signature extraction, the optimal de novo SNVs mutational signature was extracted using SigProfilerExtractor (v.1.0.18)[66]. Then, the de novo SNV extracted signature was decomposed and fitted to several SBS mutational signature datasets as the reference signatures.

During refitting, the extracted signature obtained by SigProfilerExtractor was used as the input for signature decomposition using several published SNV mutational signature sets: (1) the Catalog of Somatic Mutations in Cancer (COSMIC)[17] (https://cancer.sanger.ac.uk/signatures/sbs/) version 3.3.1 (2780 WGS from PCAWG); (2) the mutational signatures from human germline identified in the TOPMed cohort[19]; and, (3) the SNVs mutational signatures identified using KO of human DNA repair genes via targeted CRISPR-Cas9 method in isogeneic hiPSCs[22].

All signature matrix generations, decompositions, and assignments were performed using the SigProfiler suite, including R wrapper packages of SigProfilerMatrixGenerator and the Python version of SigProfilerExtractor[66]. For COSMIC mutational signatures, we used the 79 SBS signatures contained in COSMIC V3.3.1. (https://cancer.sanger.ac.uk/signatures/downloads/). For the germline-specific SNVs mutational signature, we used the 14 components of the germline mutational signature matrix data from Seplyarskiy et al.[19] (http://pklab.med.harvard.edu/ruslan/spacemut/tracks_update/TOPMed_10kb_spectra_sdnorm.txt); however, we performed some data wrangling (such as removing the non-transcribed strand) in the format of mutation types to ensure the compatibility with the SigProfiler tools. Finally, the nine SBS mutational signatures obtained from targeted CRISPR-Cas9 KO of human DNA repair/replications genes in hiPSCs were obtained from the published data by Zou et al.[22] in "Data availability" section Mutation calls (https://doi.org/10.17632/ymn3ykkmyx).

### Statistics and reproducibility

We analyzed 13 multi-child families with an average of ~4 children (range: 2 to 6 children; mean ± SD: 3.7 ± 1.2). All statistical analyses were performed in R v.4.0.2. R packages "Stats" v.4.2,1 and "Lme4" v.1.1-35.1 were used to estimate the slope confidence intervals and $p$ values of the age-related increases in SNVs. Plotting was performed with base R. Some figures were generated by Microsoft Office Professional Plus 2019 (Visio, Excel and PowerPoint).

### Reporting summary

Further information on research design is available in the Nature Portfolio Reporting Summary linked to this article.

## Code availability

The workflow used to perform signature analyses is available on github at https://github.com/hashoja/SNVs_DNMs_IRASFS and at Zenodo under https://doi.org/10.5281/zenodo.13864620.

## Data availability

Whole-genome sequencing data from the IRASFS cohort described in this study is available in the Sequence Read Archive under Bioproject access number PRJNA1166126. All 2479 validated SNVs are listed in Supplementary Data. Source data for charts/graphs presented in the main figures can be found in Supplementary data. All other data are available from the corresponding author upon reasonable request.

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

## Acknowledgements

We would like to thank: Drs. Matthew E. Hurles, Raheleh Rahbari and Sarah J. Lindsay (Wellcome Trust Sanger Institute) for advice on using DeNovoGear; Dr. Aaron R. Quinlan (University of Utah) for his help with some data analysis; Drs. Ludmil B. Alexandrov and Marcos Diaz-Gay (UC San Diego) for help and advice on using SigProfiler tools; Drs. Shamil Sunyaev and Vladimir Seplyarskiy (Harvard Medical School) for help and advice with the human germline mutational signatures; Dr. Kevin Gori (University of Cambridge) for guidance on the Sigfit mutational signature package. Finally, we would like to thank Dr. Richard Webster (Children's Hospital of Eastern Ontario) for his initial work on the REB submission and Mrs. Danielle LeBlanc (Health Canada) for helping with the sequencing contracts. Funding for this research was provided by Health Canada's Genomics Research and Development Initiative to FM. Support to CLY was provided through the Canada Research Chairs program (award number CRC-2020-00060). Grant support for IRASFS was from the National Heart, Lung and Blood Institute (NHLBI; HL060944, HL061019 and HL060919), with analysis supported by the National Institute of Diabetes and Digestive and Kidney Diseases (NIDDK; DK085175 and DK118062). The provision of whole-genome sequencing data in IRASFS Mexican Americans was supported by NHLBI (HG007112).

## Author contributions

Conceptualization: H.S., C.L.Y., F.M.; methodology, software, and data curation: H.S., M.J.M., A.S.; formal analysis: H.S., M.J.M., A.S., A.W., F.M.; investigation: H.S., A.S.; resources: N.D.P., C.L.Y., F.M.; writing–original draft: H.S., F.M.; writing–review and editing: H.S., M.J.M., A.S., A.W., N.D.P., C.L.Y., F.M.; visualization: H.S., F.M.; supervision, project administration, and funding acquisition: F.M.

## Competing interests

The authors declare no competing interests.
