## [Transparent Peer Review file · Communications Biology]

Mutational signature analyses in multi-child families reveal sources of age-related increases in human germline mutations

Corresponding Author: Dr Francesco Marchetti

Version 0:

Reviewer comments:

Reviewer #1

(Remarks to the Author)

In this manuscript, the authors examined the pattern of germline DNMs in multi-child families of Mexican-American ethnicity, identified pervasive variability in the effect of paternal age on DNMs across families, and proposed potential molecular mechanisms underlying mutagenesis through mutational signature analyses. While the study is overall legitimate, its novelty appears to be compromised by existing studies, limiting the potential impact of this paper.

1. The extensive variability in PAE has been reported in previous studies, with similar analyses like Figs 2-4 in this manuscript done on larger sample/family sizes (PMID: 31549960, 35718832). In particular, analyses as shown Fig 3 have been conducted on families with at least four children.
2. Conclusions from the mutational signature analyses largely recapitulate previous findings, with regard to the roles of 5mC deamination and DNA pathways in mutagenesis (as referenced in the manuscript [61-65]), raising the question that to what extent this study "provide a *comprehensive* picture of the mechanisms involved in the genesis of human germline" as claimed.
3. A potential novel discussion point is how these factors may differ in their effects on DNMs across families, considering the large inter-family variability seen in PAE. However, the current sample size is unlikely to support such analyses, as suggested by the authors.
4. The study applied two methods for DNM discovery and used the union, yet the DeNovoGear tool identified >4 fold more variants than GATK. It may worth further justifying the use of union rather than intersection, or applying more stringent QC on those captured by only one tool (e.g. PMID: 35718832).

In sum, I appreciate that this study adds to the population diversity of such genetic studies, yet the lack of sample sizes and informative analyses, given the existing similar, larger-scale studies, raises the concern on the novelty and significance of the messages delivered by this manuscript.

Reviewer #2

(Remarks to the Author)

In this manuscript, Shojaeisaadi et al study patterns of de novo mutations in 13 Mexican-American families. Consistent with prior literature, they identify that paternal age is a strong determinant of DNM rate. They perform mutational signature analysis to identify DNA mismatch repair / damage & replication errors as major sources of de novo variants. This paper is consistent with prior literature.

- The authors identified between 25-400 SNVs per child. This seems like a a pretty large range. Is it consistent with the literature?
- It appears that only 2,479 out of the 6,118 called SNVs were validated (line 127). What do the authors make of this low validation rate? Could this be impacting downstream analyses?
- The decrease in de novos in family 9 is an interesting observation. Is this just noise or are you powered enough to be confident in this signal? What was the p-value?
- The introduction mentions that the importance of this study is in identifying differences in ancestry-specific differences in DNM rates. However, the paper does little in the way of analyzing the mutational signatures seen in this Mexican-American cohort vs. other genetic ancestries. Without this, the novelty of the paper is limited. For example, could the authors comment on the effect size of PAE in this cohort vs. others?
- Some phenotypic data would be helpful. Are all of the families included relatively healthy? I.e., devoid of severe Mendelian

disease?

- The authors make the point that environmental exposures can also impact DNMs. Some epidemiological data on this cohort would be helpful. Where were these families recruited? How long had they been in the US? Etc.

Author Rebuttal letter:

Responses to reviewers' comments

COMMSBIO-24-0213

Shojaeisaadi et al – Mutational signature analyses in multi-child families suggest a key role for DNA mismatch repair in human germline de novo mutations.

Reviewer #1 (Remarks to the Author):

In this manuscript, the authors examined the pattern of germline DNMs in multi-child families of Mexican-American ethnicity, identified pervasive variability in the effect of paternal age on DNMs across families, and proposed potential molecular mechanisms underlying mutagenesis through mutational signature analyses. While the study is overall legitimate, its novelty appears to be compromised by existing studies, limiting the potential impact of this paper. We respectfully disagree with the reviewer and, in our responses below, we point out the novelty that distinguishes our study from the existing studies and the new findings that have originated from our approach.

1. The extensive variability in PAE has been reported in previous studies, with similar analyses like Figs 2-4 in this manuscript done on larger sample/family sizes (PMID: 31549960, 35718832). In particular, analyses as shown Fig 3 have been conducted on families with at least four children.

The reviewer is correct that similar analyses regarding the variability in the paternal age effect (PAE) have been reported in the studies of Sasani et al. and Kohalian et al. However, neither study applied the mutational signature analyses as we have done in Figure 6, nor separated out the youngest and oldest children to highlight an age-related increase in C>T mutations at CpG sites as shown in Figure 5. We note that the extensive variability in the PAE that we observe in our cohort, similarly to the previous multi-child studies in different ethnicities, supports the subsequent application of the detailed mutational signature analyses on our dataset of de novo SNVs to elucidate their mechanism of formation.

2. Conclusions from the mutational signature analyses largely recapitulate previous findings, with regard to the roles of 5mC deamination and DNA pathways in mutagenesis (as referenced in the manuscript [61-65]), raising the question that to what extent this study “provide a *comprehensive* picture of the mechanisms involved in the genesis of human germline” as claimed.

The novelty of our study is the application of three mutational signature databases to identify signatures that explain the observed de novo SNV spectrum. A few studies (e.g., Kaplanis et al.) have used mutational signature analyses of de novo SNVs obtained from family trios, but they used COSMIC signatures only to reconstruct the observed mutation spectrum.

Furthermore, Kaplanis et al. conducted mutational signature analyses only on those trios with an hypermutator phenotype caused by pre-conceptional paternal exposure to chemotherapy or because of DNA repair defects.

1

Secondly, COSMIC signatures are specific to processes present in human cancers and may not be representative of processes occurring in the germline. We report the first application of the germline-specific signatures and the DNA repair deficient mutational signatures to elucidate the molecular mechanisms potentially involved in the formation of de novo SNVs. Although some of the conclusions from these signature analyses recapitulate previous findings (e.g., the role of 5mC deamination in mutagenesis), they also provided new insight. For example, the use of the germline signatures suggests an important role for DNA damage in the genesis of SNVs; while the use of DNA repair knockouts signatures further defined a key role for PMS1, PMS2 and EXO1 in the germline mutagenesis process (see lines 332-333 of the revised manuscript).

We have added the following paragraph in the introduction (lines 91-98 of the revised manuscript) to further highlight the novel aspects of our study: “An important role for DNA mismatch repair (MMR) in the genesis of human DNMs has been inferred from the high occurrence of mutations at CpG sites due to spontaneous deamination of 5-methylcytosine

(5mC)^{20,21}. However, mechanistic studies investigating which steps of the MMR pathway are more critical for the formation of DNMs are lacking. Recently, targeted CRISPR-Cas9-based knockouts (KO) of DNA repair genes in isogenic human induced pluripotent stem cells (hiPSCs) cell lines have generated a dataset of nine DNA repair-deficient mutational signatures, including six MMR genes²². This provides an opportunity to better define the critical steps within the MMR pathway that are involved in the genesis of human DNMs.”

3. A potential novel discussion point is how these factors may differ in their effects on DMNs across families, considering the large inter-family variability seen in PAE. However, the current sample size is unlikely to support such analyses, as suggested by the authors. We have done what the reviewer is suggesting; however, because of the low cosine values obtained with the reconstruction signatures of individual families, we did not include them in the original submission. However, we have added the following text in the discussion (lines 366-374 of the revised manuscript): “We attempted to examine whether mutational signature differences among the IRASFS families contributed to the observed variability in the PAE. These analyses did show some inter-family differences especially when using the germline and DNA repair KO signatures; however, the reconstructed signatures had average cosine similarity values among the 13 families of 0.892, 0.775, and 0.809 for COSMIC, germline and DNA repair KOs, respectively (Supplementary Table 2). Overall, 46% of cosine values obtained when using the germline and DNA repair KO signatures were below the threshold (i.e., 0.8) that is considered to occur purely by chance⁸¹, which greatly reduced the confidence in the observed differences.”

4. The study applied two methods for DNM discovery and used the union, yet the DeNovoGear tool identified >4 fold more variants than GATK. It may worth further justifying the use of union rather than intersection, or applying more stringent QC on those captured by only one tool (e.g. PMID: 35718832). As indicated in the original manuscript, we used two distinct variant calling software tools to maximize the identification of candidate SNVs and ensure that we did not miss any true

2
mutation. Using multiple SNV calling algorithms has been a common approach in de novo SNV discovery to ensure that all possible variants are identified in the first computational screening step, and then subsequently validated experimentally. Thus, our choice to use the union of the two approaches as the starting list to generate baits for re-sequencing. It is worth nothing that if we use the intersection of the two approaches as the reviewer is suggesting, we obtain a set of 2,542 SNVs, which is very close to the number of validated SNVs (i.e., 2,479). This set of 2,542 SNVs also showed a paternal age effect and inter-family variability in the slope of the increase. However, not all these mutations could be re-sequenced and not all those that could be re-sequenced were validated. Thus, we have not conducted mutational signature analyses on this dataset. We have added the following text in lines 150-153 of the revised manuscript: “The set of 2,542 candidate SNVs that were identified by both DeNovoGear and GATK also showed a consistent PAE and inter-family variability in the slope of increase (data not shown). However, since not all these candidate SNVs could be re-sequenced and validated, separate data analyses on this dataset are not presented.”

In sum, I appreciate that this study adds to the population diversity of such genetic studies, yet the lack of sample sizes and informative analyses, given the existing similar, larger-scale studies, raises the concern on the novelty and significance of the messages delivered by this manuscript.

We trust that the changes we have implemented in response to the comments from this referee and referee #2 have satisfactorily addressed the expressed concern on the novelty and significance of the data presented in our manuscript.

Reviewer #2 (Remarks to the Author):

In this manuscript, Shojaeisaadi et al study patterns of de novo mutations in 13 Mexican-American families. Consistent with prior literature, they identify that paternal age is a strong determinant of DNM rate. They perform mutational signature analysis to identify DNA mismatch repair / damage & replication errors as major sources of de novo variants. This paper is consistent with prior literature.

- The authors identified between 25-400 SNVs per child. This seems like a pretty large range. Is it consistent with the literature?

We thank the reviewer for the comment. We realized that our original text was not clear, and we have revised the beginning of the paragraph to say (see lines 125-127 of the revised

manuscript): “We used two distinct variant calling software tools to maximize the identification of candidate SNVs (Supplementary Figure S1): DeNovoGear and GATK. DeNovoGear identified 123-387 candidate SNVs per child (average: 237.6); while GATK identified 24-111 candidate SNVs per child (average: 57.2). In total, 11,403 and 2,729 SNVs were identified by DeNovoGear and GATK, respectively.”

3

- It appears that only 2,479 out of the 6,118 called SNVs were validated (line 127). What do the authors make of this low validation rate? Could this be impacting downstream analyses? The number of candidate SNVs is highly dependent on the set of criteria used to analyze the sequencing data. Our criteria are presented in detail in the section entitled “Identification of candidate de novo SNVs, quality control and filtering” on pages 16-17 of the revised manuscript. The stochastic nature of high-throughput sequencing means that some areas receive high coverage while some receive lower coverage; therefore, sometimes the variant calls will be based on statistics that have few supporting reads. When the number of reads is very small, it is nearly impossible to distinguish reliably between germline and de novo variants. Thus, it is better to be permissive at the early computational steps to make sure we are as inclusive of these rare events as possible, yet experimentally validating all of them to ensure the calls are accurate. There is a balance between the cost of performing whole genome sequencing to a higher depth for the entire cohort vs. a requirement to later validate more de novo mutations. While our criteria may have resulted in overestimating the number of candidate SNVs, and thus, the low validation rate, the important point is that the number of validated SNVs in each child are aligned with the expected numbers based on the father’s age at conception (Supplementary Figure 3). Thus, we are confident that the low validation rate did not impact the downstream analyses.

- The decrease in de novos in family 9 is an interesting observation. Is this just noise or are you powered enough to be confident in this signal? What was the p-value? Concerning family 9, we had stated in the result section of the original manuscript: “This apparently unexpected observation is consistent with the range of SNVs observed in fathers of similar age (Supplementary Figure S3) and supports the role of stochastic factors in the SNV mutation rate and PAE variation 23”. However, given the reviewer’s comment, we have moved the above statement to the discussion and expanded on our interpretation of this observation. The following text has been added on lines 257-267 of the revised manuscript: “We report the first instance of a family (IRASFS Family 09) with a negative slope of -1.88 for the PAE. We believe that this apparently unexpected observation is not indicative of a fundamental biological difference in this father but is a consequence of his young age (22.3 years old at the time of the fourth child’s birth). In fact, the number of validated SNVs observed in the four children are consistent with the range of SNVs observed in fathers of similar age (Supplementary Figure S3). Secondly, both previous multi-family studies^{14, 15} have several cases with downward trends in the number of SNVs when limited to the first few children. It is only because these families have children fathered at an older age that the slope turned positive. Thus, we hypothesize that if IRASFS Family 09 had a fifth or sixth child, this negative slope would disappear. Overall, these data support the role of stochastic factors in the SNV mutation rate and PAE variation 23.”

- The introduction mentions that the importance of this study is in identifying differences in ancestry-specific differences in DNM rates. However, the paper does little in the way of analyzing the mutational signatures seen in this Mexican-American cohort vs. other genetic

4

ancestries. Without this, the novelty of the paper is limited. For example, could the authors comment on the effect size of PAE in this cohort vs. others?

We thank the reviewer for the comment. We have created a new Supplementary Figure 4 that shows the effect size of the paternal age effect in all multi-child families from the three studies (Sasani et al, Kohalian et al, and our study) sorted by the increasing slope of the number of SNVs per year of paternal age. This new figure shows that families from the three different ancestries are randomly distributed. We have added the following text in lines 249-252 of the revised manuscript: “In addition, when the multi-child families from these two previous studies and ours are sorted based on the increasing slope of the PAE, we observed a random distribution of the families, irrespectively of ethnicity (Supplementary Figure 4)”. This provides additional support for our original conclusion (lines 252-253 of the revised manuscript) that: “Therefore, it appears that the PAE and its inter-family variability is a general characteristic of the human species that is independent of ethnicity.”

- Some phenotypic data would be helpful. Are all of the families included relatively healthy? I.e., devoid of severe Mendelian disease?

In response to the referee's comment, we have added a section in the Materials and Methods to include a more extensive description of the cohort. The following text added on lines 412-422 of the revised manuscript: "Study Cohort. The IRASFS cohort is a population-based cohort designed to investigate the genetic and epidemiologic basis of glucose homeostasis and abdominal adiposity with a focus on Mexican-derived participants. The study design, recruitment methods and phenotype assessment have been described previously²³. Broadly, Mexican American families were recruited from two clinical centers including San Antonio, TX, and San Luis Valley, CO, in 1999-2002 as an extension of the original IRAS cohort recruited in 1992-1994²³. The overall cohort was relatively healthy and devoid of severe Mendelian diseases. Individual-level genetic data and ADMIXTURE analysis indicated homogeneity across the cohort^{74,75}. Specific to the 13 families studied here, and mirroring the larger cohort^{76,77}, subjects were mostly female (64.6%) with an average age of 47 years, overweight (27.20 kg/m²) and with a near optimal lipid levels (104.24 mg/dL)."

- The authors make the point that environmental exposures can also impact DNMs. Some epidemiological data on this cohort would be helpful. Where were these families recruited? How long had they been in the US? Etc.
See our response to the comment above.

5

Version 1:

Reviewer comments:

Reviewer #1

(Remarks to the Author)

The authors have defended the novelty of their work well - the manuscript is now clearer and more convincing on this point.

Reviewer #2

(Remarks to the Author)

the authors have sufficiently addressed my comments.
